

# A mixed finite element discretisation of the shallow water equations

James Kent[1], Thomas Melvin[1], and Golo Wimmer[2]

[1]Dynamics Research, Met Office, Exeter, UK
[2]Los Alamos National Laboratory, Los Alamos, New Mexico 87545, USA

**Correspondence:** James Kent, Met Office, FitzRoy Road, Exeter EX1 3PB, UK. (james.kent@metoffice.gov.uk)

**Abstract.** This paper introduces a mixed finite-element shallow water model on the sphere. The mixed finite-element approach is used as it has been shown to be both accurate and highly scalable for parallel architecture. Key features of the model are an iterated semi-implicit time stepping scheme, a finite-volume transport scheme, and the cubed sphere grid. The model is tested on a number of standard spherical shallow water test cases. Results show that the model produces similar results to other
shallow water models in the literature.

## 1 Introduction

The dynamical core of an atmospheric model numerically approximates the solution to the governing fluid dynamics equations that determine the evolution of the atmosphere. An operational dynamical core must be both accurate, to give confidence in the forecast, and efficient, to produce a forecast in the desired time frame. Current supercomputer architectures focus on using an

increasing number of processors to decrease runtime, and so future dynamical cores must be suitable for parallel architecture. Key aspects of the dynamical core that will greatly affect both accuracy and parallel scalability are the type of grid and the numerical methods used to discretise the equations.

Traditionally, atmospheric models used a latitude-longitude grid (Wood et al., 2014). However, the convergence of the meridians at the pole leads to a computational bottleneck, and thus the latitude-longitude grid is not suitable for future super-

computers. Hence, a number of different approaches to gridding the sphere without a pole have been developed and used in dynamical cores, such as icosahedral grids, and the yin-yang grid (Staniforth and Thuburn, 2012). One such promising grid is the cubed sphere, which uses quadrilateral cells on six panels that cover the sphere (Ronchi et al., 1996).

The choice of numerical methods used in a dynamical core will affect both the accuracy and the efficiency of the model. High-order methods generally improve accuracy but increase the computational cost. There are a variety of spatial methods





(for example, finite-difference, finite-volume, semi-Lagrangian) and temporal methods (for example explicit, semi-implicit, horizontally-explicit vertically-implicit) used in atmospheric models by different operational centres and modelling groups (see Ullrich et al. (2017) and references within). The mixed finite-element method of Cotter and Shipton (2012) and Cotter and Thuburn (2014) is a spatial method that allows high-order schemes to be defined whilst retaining good parallel properties (Melvin et al., 2019). In practice, most current atmospheric dynamical cores aim for second-order accuracy overall (with the

transport scheme often higher-order) (Wood et al., 2014). For this reason the lowest-order mixed finite-element method will be a building block of the model described in this article.

The shallow water equations are an important step in testing the methods used for the development of a dynamical core (Williamson et al., 1992). The shallow water equations contain many of the properties of the full atmospheric governing equations, but with a reduction in complexity. In practice, if a method is not *good enough* (by whatever metric) for a shallow

water model, it will not be *good enough* for an atmospheric dynamical core. This allows the methods to be evaluated in a simpler setting, and design decisions to be made early in development.

This document presents a novel mixed finite element discretisation, extending that of Melvin et al. (2019) for atmospheric dynamics in Cartesian geometry, of the rotating shallow water equations in a spherical domain. This is the natural next step from Melvin et al. (2019) in the development of a 3D dynamical core in spherical geometry, allowing the impact of the chosen

spherical grid to be investigated in a simplified context using an evolution of the same numerics. As with the model of Melvin et al. (2019), the shallow water discretisation presented here has been developed in the LFRic framework, see Adams et al. (2019). Key components of the model are the semi-implicit iterated time stepping scheme and the finite-volume transport. The governing shallow water equations are given in Section 2. The discretisation, including the spatial and temporal aspects, is given in Section 3, the finite-volume transport scheme is described in Section 4, and the solution procedure is outlined in

Section 5. Results from standard shallow water test suites are given in 7, with a concluding summary in Section 8.

## 2   Governing Equations

The shallow water equations in a rotating domain are given in vector-invariant form by

$$\frac{\partial \mathbf{u}}{\partial t} + q\Phi\mathbf{u}^{\perp} + \nabla\left(K + \Phi + \Phi_S\right) = 0, \tag{1}$$

$$\frac{\partial \Phi}{\partial t} + \nabla\cdot(\mathbf{u}\Phi) = 0, \tag{2}$$

where $\mathbf{u} = (u,v)$ is the velocity, $\Phi = gh$ is the geopotential (with gravity $g$ and free surface height $h$), $\Phi_s$ is the surface geopotential and $K \equiv 1/2|\mathbf{u}|^2$ is the kinetic energy. The perpendicular operator is defined by $(u,v)^{\perp} = (-v,u)$.

The potential vorticity (PV) $q$ is defined as

$$q = \frac{\nabla^{\perp}\cdot\mathbf{u} + f}{\Phi}, \tag{3}$$

where $f$ is the Coriolis parameter and $\nabla^{\perp}\cdot \equiv \mathbf{k}\cdot\nabla\times$, and taking the curl of (1) gives a conservative transport equation for PV

$$\frac{\partial \Phi q}{\partial t} + \nabla\cdot(\mathbf{u}\Phi q) = 0. \tag{4}$$





## 3  Discretisation

The governing equations (1) and (2) are discretised in time using a two-time level iterated implicit scheme (Section 3.1) and in space using a mixed finite element method (Section 3.2) for the wave-dynamic terms and a high order upwind finite volume scheme (Section 4) for the advection terms. This process is an extension to the shallow water equations on the sphere of Melvin

et al. (2019) who presented a similar discretisation in a 3D Cartesian domain and the interested reader is referred there for more information.

### 3.1  Temporal Discretisation

To achieve second-order temporal accuracy a time centred approach is used. The target discretisation, as in Melvin et al. (2019), is a two-time-level iterated implicit scheme where terms responsible for the fast wave dynamics are treated by the iterative semi-

implicit scheme and the transport terms (those involving the mass flux in (2) and the potential vorticity in (1)) are computed using a high-order, upwind, explicit finite volume scheme. Taking equations (1) and (2) and applying this discretisation results in

$$\delta_t \mathbf{u} + \boldsymbol{\mathcal{F}}^\perp \left( q^n \Phi^n, \overline{\mathbf{u}}^{1/2} \right) + \nabla \overline{(K + \Phi + \Phi_s)}^\alpha \;\; = \;\; 0, \tag{5}$$

$$\delta_t \Phi + \nabla \cdot \boldsymbol{\mathcal{F}} \left( \Phi^n, \overline{\mathbf{u}}^{1/2} \right) \;\; = \;\; 0 \tag{6}$$

where $\delta_t s \equiv \left( s^{n+1} - s^n \right) / \Delta t$, $\overline{s}^\alpha \equiv \alpha s^{n+1} + (1-\alpha) s^n$ and $\boldsymbol{\mathcal{F}}(s, \mathbf{u})$ is the flux computed by the transport scheme of variable $s$ by wind field $\mathbf{u}$. We use $\alpha = 1/2$ to achieve the second-order centred in time scheme.

### 3.2  Mixed Finite-Element discretisation

The mixed finite element formulation in two spatial dimensions requires the specification of three finite element function spaces: $\mathbb{V}_0$, $\mathbb{V}_1$, $\mathbb{V}_2$ (c.f. the four function spaces $\mathbb{W}_i$, $i = 0...3$ used in Melvin et al. (2019)). The scalar spaces are an $H_1$ space

consisting of pointwise scalars: $\mathbb{V}_0$ (zero-forms), or a $L_2$ space consisting of area integrated scalars: $\mathbb{V}_2$ (two-forms). There are two choices for the vector space $\mathbb{V}_1$ corresponding to either $\mathbb{V}_1^C$: a $H_{\mathrm{curl}}$ space of circulation vectors or $\mathbb{V}_1^D$: a $H_{\mathrm{div}}$ space of flux vectors. Each choice has an associated discrete de-Rham complex:

$$\mathbb{V}_0 \xrightarrow{\nabla} \mathbb{V}_1^C \xrightarrow{\mathbf{k} \cdot \nabla \times} \mathbb{V}_2, \tag{7}$$

for curl conforming vectors, and

$$\mathbb{V}_0 \xrightarrow{\mathbf{k} \times \nabla} \mathbb{V}_1^D \xrightarrow{\nabla \cdot} \mathbb{V}_2, \tag{8}$$

for div conforming vectors. For this paper only the div conforming complex (8) will be considered $\mathbb{V}_1 \equiv \mathbb{V}_1^D$, with $\mathbf{u} \in \mathbb{V}_1$ and $\Phi \in \mathbb{V}_2$. This is analogous to the standard C-grid staggering (Arakawa and Lamb, 1977) where the normal components of the velocity vector are stored at the cell edges.

The potential vorticity $q \in \mathbb{V}_2$ and is computed diagnostically as follows. The velocity $\mathbf{u}$ is represented in the $H_{\mathrm{curl}}$ space

as $\mathbf{v} \in \mathbb{V}_1^C$ using a Galerkin projection, then the curl is taken to give the relative vorticity $\omega \equiv \mathbf{k} \cdot \nabla \times \mathbf{v} \in \mathbb{V}_2$. The absolute





vorticity is projected into $\mathbb{V}_2$, then divided by the geopotential. As the lowest order space is used this gives the potential vorticity $q \in \mathbb{V}_2$.

Taking (5), multiplying by a test function $\mathbf{w}$ from the velocity space and integrating over the domain D gives

$$\int_D \mathbf{w} \cdot \left[ \delta_t \mathbf{u} + \boldsymbol{\mathcal{F}}^\perp \left( q^n \Phi^n, \overline{\mathbf{u}}^{1/2} \right) \right] - \nabla \cdot \mathbf{w} \overline{\left( K + \Phi + \Phi_s \right)}^\alpha dA = 0, \tag{9}$$

where, since the geopotential and kinetic energy are discontinuous between cells the third term has been integrated by parts and the boundary term vanishes due to the continuity of the test functions $\mathbf{w}$. Similarly taking (6), multiplying by a test function $\sigma$ from the geopotential space and integrating over the domain gives

$$\int_D \sigma \left[ \delta_t \Phi + \nabla \cdot \boldsymbol{\mathcal{F}} \left( \Phi^n, \overline{\mathbf{u}}^{1/2} \right) \right] dA = 0, \tag{10}$$

where it is assumed that the advection scheme returns fluxes $\boldsymbol{\mathcal{F}} \in \mathbb{V}_1$.

## 3.3 Transforms

As in Melvin et al. (2019), the equations are transformed from a physical cell C to a reference cell $\widehat{\text{C}}$ using the mapping $\phi : \widehat{\text{C}} \to \text{C}$. The physical cell, on the cubed sphere, has coordinates $\boldsymbol{\chi}$ and the reference cell, a unit square, has coordinates $\widehat{\boldsymbol{\chi}}$, and the transform is such that $\boldsymbol{\chi} = \phi(\widehat{\boldsymbol{\chi}})$. Transforming the equations to a single reference cell provides a number of computational efficiencies such as a single set of basis functions and quadrature points (Rognes et al., 2009). The Jacobian of

this transformation is defined as $\mathbf{J} \equiv \partial \phi (\widehat{\boldsymbol{\chi}}) / \partial \widehat{\boldsymbol{\chi}}$, and is used in transforming variables between the physical and reference cells. The transformations used for spaces $\mathbb{V}_1$ and $\mathbb{V}_2$ are designed to preserve fluxes through an edge $(\mathbb{V}_1)$ and area integrated values $(\mathbb{V}_2)$ respectively. The transformation for $\mathbb{V}_1$ is $\mathbf{v}(\boldsymbol{\chi}) \equiv \mathbf{v}(\phi[\widehat{\boldsymbol{\chi}}]) = \mathbf{J}\widehat{\mathbf{v}}(\widehat{\boldsymbol{\chi}}) / \det \mathbf{J}$. Following Melvin et al. (2019), for the $\mathbb{V}_2$ transformation rehabilitation (Bochev and Ridzal, 2010) is used so that the $\mathbb{V}_2$ mapping is modified to $\sigma(\boldsymbol{\chi}) \equiv \sigma(\phi[\widehat{\boldsymbol{\chi}}]) = \widehat{\sigma}(\widehat{\boldsymbol{\chi}})$. Applying these to (9) and (10) gives

$$\int_D \frac{\mathbf{J}\widehat{\mathbf{w}}}{\det \mathbf{J}} \cdot \left[ \delta_t \mathbf{J}\widehat{\mathbf{u}} + \mathbf{J}\widehat{\boldsymbol{\mathcal{F}}}^\perp \left( q^n \Phi^n, \overline{\mathbf{u}}^{1/2} \right) \right] - \widehat{\nabla} \cdot \widehat{\mathbf{w}} \overline{\left( \widehat{K} + \widehat{\Phi} + \widehat{\Phi_s} \right)}^\alpha d\widehat{A} = 0, \tag{11}$$

and

$$\int_D \widehat{\sigma} \left[ \delta_t \widehat{\Phi} \det \mathbf{J} + \widehat{\nabla} \cdot \widehat{\boldsymbol{\mathcal{F}}} \left( \Phi^n, \overline{\mathbf{u}}^{1/2} \right) \right] d\widehat{A} = 0. \tag{12}$$

## 4 Transport scheme

The transport scheme is an extension to the method of lines scheme used by Melvin et al. (2019), computing fluxes $\boldsymbol{\mathcal{F}}$ of a
scalar field $s$ by a wind field $\mathbf{u}$

$$\boldsymbol{\mathcal{F}}(s, \mathbf{u}) = \int_t^{t+\Delta t} s\mathbf{u} \, dt. \tag{13}$$





The flux $\mathcal{F}$ is obtained using a method of lines scheme where a conservative transport equation

$$s^{n+1} - s^n + \Delta t \nabla \cdot \mathcal{F}(s^n, \mathbf{u}) = 0, \tag{14}$$

is solved to obtain $\mathcal{F}$. The temporal aspects of this scheme are handled in the same manner as Melvin et al. (2019) using a
$m$-stage Runge-Kutta scheme

$$s^{(i)} = s^n - \Delta t \sum_{j=1}^{i-1} a_{i,j} \nabla \cdot \mathbf{F}\left(s^{(j)}, \mathbf{u}\right),$$

$$i = 1, \ldots, m, \tag{15}$$

$$s^{n+1} = s^n - \Delta t \sum_{k=1}^{m} b_k \nabla \cdot \mathbf{F}\left(s^{(k)}, \mathbf{u}\right), \tag{16}$$

the coefficients $a_{i,j}$ and $b_k$ in (15) and (16) are given by the Butcher tableau for the scheme


$$
\begin{array}{c|ccccc}
0 & & & & & \\
c_2 & a_{2,1} & & & & \\
c_3 & a_{3,1} & a_{3,2} & & & \\
\vdots & \vdots & & \ddots & & \\
c_m & a_{m,1} & \cdots & & a_{m,m-1} & \\
\hline
& b_1 & b_2 & \cdots & b_{m-1} & b_m
\end{array}
$$

Here the 3-stage 3rd-order strong stability preserving (SSP3) method of Gottleib (2005) is used which has the Butcher tableau

$$
\begin{array}{c|ccc}
0 & & & \\
1 & 1 & & \\
1/2 & 1/4 & 1/4 & \\
\hline
& 1/6 & 1/6 & 4/6
\end{array}
$$

At each stage $i$ we need to compute $\mathbf{F}\left(s^{(i)}, \mathbf{u}\right) \equiv \check{s}^{(i)} \mathbf{u}$ where $\check{s}$ is a high order upwind reconstruction of $s$. Here the spatial reconstruction of Melvin et al. (2019) is extended to take into account the nonuniformity of the mesh by using a two dimensional
horizontal reconstruction. The scheme defined in this section works on order $l = 0$ spaces.

### 4.1  Reconstruction of a scalar field

The advection scheme computes a high order upwind reconstruction $\check{s}$ of a given scalar field $s$. The reconstructed field is computed at points staggered half a grid length in all directions from the original field, so for a field $s \in \mathbb{V}_2$, which is located at cell centres, then the reconstructed field $\check{s}$ is computed at the centre of each cell edge. The reconstruction is computed by fitting
a polynomial through a number of cells and evaluating this polynomial at the staggered points. This is given an upwind bias by choosing even order polynomials for the reconstruction which require an odd number of $s$ points and hence can be weighted





to the upwind side of the point at which the reconstruction is needed, e.g for a one-dimensional quadratic reconstruction at a point $\check{s}_{i+1/2}$ with a positive wind field, two upwind points $s_{i-1}$ and $s_i$ and one downwind point $s_{i+1}$ are used.

The horizontal spatial reconstruction is based on that used in Thuburn et al. (2014) and the interested reader is referred
to Baldauf (2008) and Skamarock and Menchaca (2010) for other results on these types of schemes. To summarise, a series of polynomials $P_k$ of a given order $n$ in a local Cartesian coordinate system $(x,y)$ is defined over a stencil of $n_s$ cells. The polynomial is required to fit (in a least squares sense) the discrete field being reconstructed. The integral along the cell edge of the reconstructed field $\check{s}$ is approximated by Gaussian quadrature and given by

$$\int \check{s}\,dS \equiv \int \sum_{k=1}^{n_s} P_k(x,y)\,s_k\,dS \approx \sum_{j=1}^{n_q}\sum_{k=1}^{n_s} w_j P_k(x_j,y_j)\,s_k, \tag{17}$$

where $(x_j,y_j)$ are the integration points and $w_j$ the weights of the Gaussian quadrature. In practice $n_q = 2$ point quadrature is used and this is found to give a small improvement over single point quadrature.

The weights $P_k(x_r, y_r)$ that multiply each value $s_k$ in the stencil are obtained by evaluating a polynomial

$$P_k(x,y) = \sum_{i=0}^{n}\sum_{j=0}^{n-i} a_{i,j}^k x^i y^j, \tag{18}$$

at $(x_r, y_r)$. The coefficients $a_{i,j}^k$ of $P_k$ are determined by minimising the residual $r_k$

$$r_k = \sum_{j=1}^{ns}\left[\int_j P_k - \delta_{jk}\,dA_j\right]^2, \tag{19}$$

so that the integral of $P_k = 1$ in cell $k$ and $P_k = 0$ otherwise.

For an order $n$ reconstruction there are $n_m \equiv (n+1)(n+2)/2$ coefficients $a_{i,j}^k$ and so to avoid an under-determined problem this requires at least $n_m$ cells in the stencil. Additionally the stencil should be symmetric about the central cell. To ensure these properties hold, the stencils are generated in the same manner as Thuburn et al. (2014), to summarise the following algorithm
is used:

1. Add the central cell to the stencil (if $n = 0$, stop).

2. Loop until number of cells in the stencil $n_s$ is at least the number of monomials $n_m$.

3. Find the set $S$ of all neighbouring cells of cells currently in the stencil.

4. *Either* add all cells in $S$ that are not already in the stencil and are neighbours of 2 cells already in the stencil, *or* if no cell
in $S$ is a neighbour of two cells in the stencil then add all cells in $S$ that aren't already in the stencil.

An example of the type of stencil this generates around a corner of the cubed sphere is shown in Figure 1.

For example with a quadratic reconstruction $n = 2$ there are $n_m = 6$ monomials and the stencil algorithm will generate a stencil with $n_s = 9$ cells in general and $n_s = 8$ cells near the corners of the cubed sphere. As in Thuburn et al. (2014) the central cell ($k = 1$) in the stencil is fitted exactly ($r_1 = 0$ in (19)) and the others are fitted in a least squares sense.





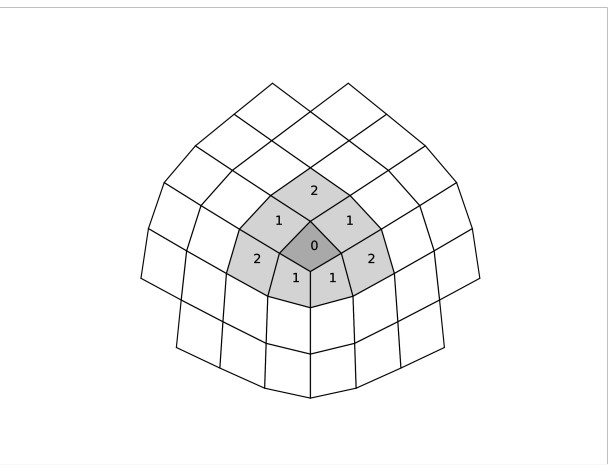

**Figure 1.** Stencil for a quadratic reconstruction of a field around the corner of a cubed-sphere. Numbers indicate which iteration of the stencil generation the cell is added at, lightly shaded cells indicate cells used for the reconstruction that are fitted in a least squares manner and the darkly shaded cell indicates the central cell which is fitted exactly.

The local Cartesian coordinates $(x, y)$ are computed as in Thuburn et al. (2014), the origin of the coordinate system $\mathbf{x}_0$ is taken to be the centre of the cell in the centre of the stencil and the direction of the $x-$axis is then taken as the direction from $\mathbf{x}_0$ to an arbitrary neighbour. Then it is straightforward to reconstruct any point $\mathbf{x}_i \equiv (X_i, Y_i, Z_i)$ in the stencil in terms of the local coordinates $(x_i, y_i)$, see Thuburn et al. (2014) for more details.

### 4.2 Flux computation

Once the scalar field $s$ has been reconstructed at the locations of the $\mathbb{V}_1$ degrees of freedom to give $\check{s}$, the flux of that field on a cell face is obtained by multiplying the reconstructed scalar by the normal component of the velocity field at that point

$$\widehat{\mathcal{F}} = \widehat{\mathbf{u}}\check{s},$$

(20)





where $\check{\mathbf{s}}$ is the vector containing all $\check{s}$ at cell edges.

### 4.3 Predictors

The temporal discretisation is designed to mimic a semi-implicit semi-Lagrangian discretisation such as that used in Wood et al. (2014). For the transport scheme this includes advecting predictors $(\Phi^p, q^p)$ for the geopotential and potential vorticity fields instead of the values at the start of the timestep $(\Phi^n, q^n)$. This is motivated by considering a semi-Lagrangian discretisation of a prototypical equation

$$\frac{Ds}{Dt} + G(s) = 0, \tag{21}$$

with $D/Dt \equiv \partial/\partial t + \mathbf{u} \cdot \nabla$. Discretising across a trajectory from $\mathbf{x}_D$ at time level $n$ to $\mathbf{x}_A$ at time level $n+1$ gives

$$[s + \alpha \Delta t G(s)]_A^{n+1} = [s - (1-\alpha)\Delta t G(s)]_D^n \tag{22}$$

with subscripts $A$ and $D$ denoting evaluation at arrival $\mathbf{x}_A$ and departure $\mathbf{x}_D$ points respectively. Evaluation of a function $F$ at a departure point can be expressed as

$$F_D^n = F_A^n - \mathcal{A}_{sl}\left(F^n, \overline{\mathbf{u}}^{1/2}\right) \tag{23}$$

where $\mathcal{A}_{sl}$ is the operation of the semi-Lagrangian advection operator. Applying this approximation to (21) gives

$$[s + \alpha \Delta t G(s)]^{n+1} = s^p - \mathcal{A}_{sl}\left[s^p, \overline{\mathbf{u}}^{1/2}\right], \tag{24}$$

$$s^p \equiv [s - (1-\alpha)\Delta t G(s)]^n \tag{25}$$

where the subscript $A$ has been dropped for convenience. Applying this idea to (2) the geopotential predictor to be advected is then

$$\Phi^p \equiv [\Phi - (1-\alpha)\Delta t \Phi \nabla \cdot \mathbf{u}]^n. \tag{26}$$

This can also be applied to potential vorticity using (4). These predictors come from considering the continuity and PV equations in advective form $Ds/Dt + s\nabla \cdot \mathbf{u} = 0$ where $s = \Phi$ or $q\Phi$ as would be used in a semi-Lagrangian model.

## 5 Solution Procedure

The procedure for the solution of the shallow water equations is as follows. The semi-implicit governing equation, (5) and (6),

are repeated here for clarity:

$$\delta_t \mathbf{u} + \mathcal{F}^\perp\left(q^n \Phi^n, \overline{\mathbf{u}}^{1/2}\right) + \nabla\overline{(K + \Phi + \Phi_s)}^\alpha = 0, \tag{27}$$

$$\delta_t \Phi + \nabla \cdot \mathcal{F}\left(\Phi^n, \overline{\mathbf{u}}^{1/2}\right) = 0. \tag{28}$$



An iterated implicit scheme is used to solve (27) and (28). At each stage $(k)$ of the iterative scheme the time-level $n+1$ terms are lagged and included in the residuals such that

$$\mathbf{R}_u \equiv (\mathbf{u} + \alpha\Delta t\nabla(K + \Phi + \Phi_s))^{(k)} - (\mathbf{u} - [1-\alpha]\,\Delta t\nabla(K + \Phi + \Phi_s))^n + \Delta t\boldsymbol{\mathcal{F}}^{\perp}\left((q\Phi)^p, \overline{\mathbf{u}}^{1/2}\right), \tag{29}$$

$$R_\Phi \equiv \Phi^{(k)} - \Phi^n + \Delta t\nabla\cdot\boldsymbol{\mathcal{F}}\left(\Phi^p, \overline{\mathbf{u}}^{1/2}\right), \tag{30}$$

where $k$ is the estimate for the $n+1$ terms after $k$ iterations of the iterative scheme. Increments $\mathbf{u}' \equiv \mathbf{u}^{(k+1)} - \mathbf{u}^{(k)}$ and $\Phi' \equiv \Phi^{(k+1)} - \Phi^{(k)}$ to (29)-(30) are sought such that the fast wave terms are handled implicitly

$$\mathbf{u}' + \tau f(\mathbf{u}')^{\perp} + \tau\Delta t\nabla\Phi' \quad = \quad -\mathbf{R}_u, \tag{31}$$

$$\Phi' + \tau\Delta t\nabla\cdot(\Phi^*\mathbf{u}') \quad = \quad -R_\phi, \tag{32}$$

where $\Phi^*$ is a reference state used to obtain the linearisation and $\tau$ is a relaxation parameter (usually chosen to be $\tau = 1/2$). In practice, we use $\Phi^* = \Phi^n$ as the reference state. Applying the mixed finite element discretisation this becomes the system

$$\begin{pmatrix} M_1 + \tau\Delta tC & -\tau\Delta tD\,(1\cdot)^T \\ \tau\Delta tD\,(\Phi^*\cdot) & M_2 \end{pmatrix} \begin{pmatrix} \mathbf{u}' \\ \Phi' \end{pmatrix} = -\begin{pmatrix} \mathcal{R}_u \\ \mathcal{R}_\Phi \end{pmatrix} \tag{33}$$

where $\mathcal{R}_u$ and $\mathcal{R}_\Phi$ are the finite-element discretisations of (29) and (30) respectively given by (11) and (12). The matrices are defined as

$$M_1^{i,j} \equiv \quad \int_{\mathrm{D}} \mathbf{J}\widehat{\mathbf{w}}_i\cdot\frac{\mathbf{J}}{\det\mathbf{J}}\widehat{\mathbf{w}}_j\,d\widehat{A}, \tag{34}$$

$$M_2^{i,j} \equiv \quad \int_{\mathrm{D}} \widehat{\sigma}_i\widehat{\sigma}_j\det\mathbf{J}\,d\widehat{A}, \tag{35}$$

$$C^{i,j} \equiv \quad f\int_{\mathrm{D}} \mathbf{J}\widehat{\mathbf{w}}_i\cdot\frac{\mathbf{J}}{\det\mathbf{J}}\widehat{\mathbf{w}}_j^{\perp}\,d\widehat{A}, \tag{36}$$

$$D\,(\psi)^{i,j} \equiv \quad \int_{\mathrm{D}} \widehat{\nabla}\cdot\widehat{\mathbf{w}}_i\psi\sigma_j\,d\widehat{A} \tag{37}$$

At each iterate $(k)$ the system (33) is solved using an iterative Krylov subspace method (in this case GMRES).

## 6 Mesh

An equiangular cubed sphere mesh is used to grid the sphere. We use the notation C$n$ to describe a cubed sphere with $n \times n$ cells per panel. The mesh is parameterised using a finite-element representation of the sphere within a cell with polynomials of order $m$. Note therefore, that a point within a cell does not necessarily lie on the sphere, with the error depending on the order of the elements used. To compute the error, the geocentric coordinates $(X, Y, Z)$ at a point in the cell are computed using the finite-element representation. The error is the difference between the true radius of the sphere and the radius using $(X^2 + Y^2 + Z^2)^{1/2}$.

Figure 2 shows the error within a cell when using linear and quadratic elements on a C96 grid with the Earth's radius. The error for the linear element is largest at the cell centre, with a maximum error of $426.39$ m. The quadratic element reduces the error by $5$ orders of magnitude, with a maximum error of $0.0018$ m. The right plot of Figure 2 shows the convergence of





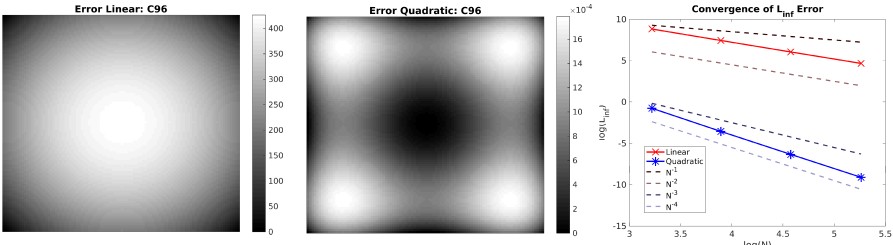

**Figure 2.** The error from linear and quadratic finite-element representations of the sphere within a cell. The left and centre plots show the absolute error (true - FE) for the linear and quadratic basis functions respectively on the $C96$ cubed sphere grid. The right plot shows the convergence of the $\ell_{\text{inf}}$ error, using $C24$, $C48$, $C96$ and $C192$ cubed sphere grids.

|  | C24 | C48 | C96 |
|---|---|---|---|
| $\ell_2(\Phi)$ | 4.86e-4 | 1.04e-4 | 2.22e-5 |
| $\ell_\infty(\Phi)$ | 6.19e-4 | 1.40e-4 | 3.17e-5 |

**Table 1.** The normalised $\ell_2$ and $\ell_\infty$ geopotential errors norms after 15 days for the Williamson 2 test at different resolutions.

the maximum error within a cell. The linear element error converges at second-order, with the quadratic element converging at fourth-order. For this reason the quadratic element is used to create the mesh for the shallow water model.

## 7  Numerical Results

This section shows the results of the model runs using a standard set of spherical shallow water test cases. The full initial
conditions for each of the tests are given in the references under each test case.

### 7.1  Williamson 2: Steady-State

The first test case is the steady-state test described in Williamson et al. (1992). The steady flow means that the initial conditions are the analytical solution at any time, and thus error norms can be calculated for the runs at different resolution.

The normalised $\ell_2$ errors norms of the geopotential field at 15 days are given in table 1 for $C24$ ($\Delta t = 3600$), $C48$ ($\Delta t =$
$1800$) and $C96$ ($\Delta t = 900$) resolutions. The error norms can be used to determine the convergence rate and hence the empirical order-of-accuracy of the model. The convergence rate of the $\ell_2$ and $\ell_\infty$ error norms are approximately second-order. This is expected as the finite-element discretisation and the timestepping scheme are both second-order methods. The error fields after 15 days for $C96$ resolution are shown in Figure 3, these errors show a wave number 4 pattern coming from the underlying cubed sphere mesh, however the errors are large scale and are not particularly clustered around the edges and corners of the
cubed sphere indicating an acceptable level of grid imprinting.





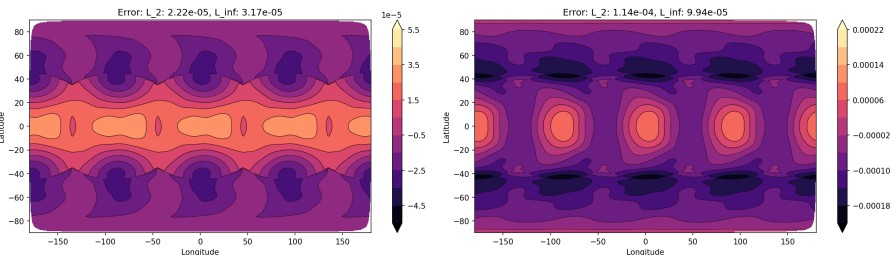

**Figure 3.** Errors after 15 days for the Williamson 2 test at C96 resolution for geopotential (left plot) and zonal wind (right plot) showing the large scale nature of the error.

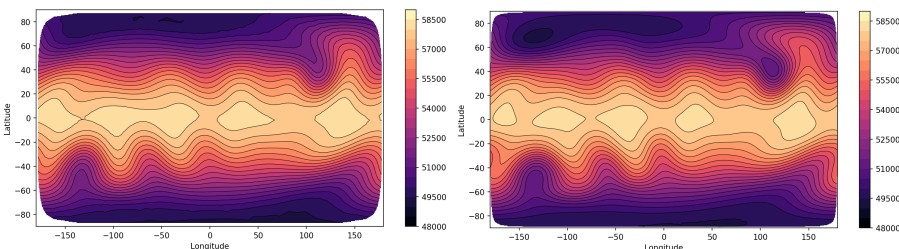

**Figure 4.** The total geopotential ($\Phi + \Phi_s$) after 15 days for the Williamson mountain test. The left plot shows the solution using $C24$ resolution and the right plot shows the $C96$ solution.

### 7.2 Williamson 5: Mountain Test

The mountain test case of Williamson et al. (1992) is used to show the performance of the model when orography is present (i.e. $\Phi_s \neq 0$). The initial zonal flow is over a mountain centred at $-90°$ longitude and $30°$ latitude.

The total geopotential at day 15 for C24 and C96 resolutions (with $\Delta t = 3600$ and $900$ s respectively) is shown in Figure 4. Both resolutions capture the features of the flow for this test. The results are comparable to other shallow water models at similar resolutions (see, for example Thuburn et al. (2010)), demonstrating the model's ability to correctly simulate flow over orography.

To demonstrate the conservation properties of the model the mass, the total energy

$$E = \int_{D} \frac{1}{2} h \left( |\mathbf{u}|^2 + \Phi + 2\Phi_s \right) dA, \tag{38}$$

and the potential enstrophy

$$Z = \int_{D} \frac{1}{2} \Phi q^2 dA, \tag{39}$$

where the integrals are over the whole domain, are computed at each time step.



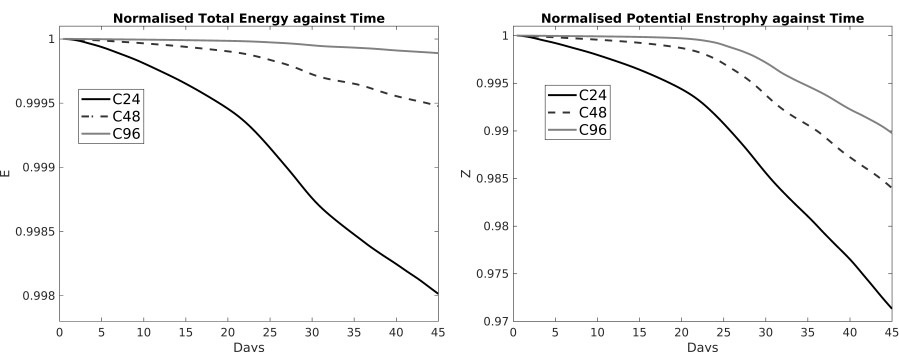

**Figure 5.** The normalised total energy (left) and potential enstrophy (right) against time for the Williamson mountain test.

Mass, energy and potential enstrophy are conserved in the continuous equations. As the model uses a finite-volume transport scheme, the mass is conserved to machine precision, however energy and potential enstrophy are not conserved by the dis-
cretisation presented here and in fact, in the discrete case, potential enstrophy cascades downscale from resolved to unresolved scales, and thus is expected to decrease with time.

The normalised total energy and potential enstrophy are plotted against time in Figure 5 for different grid resolutions (C24, C48, and C96) up to day 45. As the resolution increases the dissipation in the model decreases, and the total energy and potential enstrophy curves are closer to conservation (a horizontal line). The flow is initially weakly non-linear, and so for the
first 15 days there are not significant cascades to unresolved scales. After 15 days the percentage loss in total energy is $0.0349\%$ for C24, $0.0062\%$ for C48, and $0.001\%$ for C96. For potential enstrophy the percentage loss is $0.358\%$ for C24, $0.076\%$ for C48, and $0.014\%$ for C96. Extending to 45 days the flow becomes more non-linear, and this results in more dissipation of energy and potential enstrophy. The percentage losses are $0.199\%$ for C24, $0.052\%$ for C48, and $0.011\%$ for C96 in the total energy, and $2.87\%$ for C24, $1.60\%$ for C48, and $1.02\%$ for C96 in the potential enstrophy.

### 260 7.3 Galewsky Instability Test

For the Galewsky instability test (Galewsky et al., 2004) a perturbation is added to a balanced jet to create a barotropic instability. As the instability progresses many small scale potential vorticity filaments are produced.

The potential vorticity solutions at day 6 are shown in Figure 6 for the $C48$, $C96$, $C192$ and $C384$ resolution runs (with $\Delta t = 900, 450, 225,$ and $112.5$ s respectively). At $C48$ resolution the grid imprinting from the cubed sphere is evident. A wave
number 4 pattern appears on the jet, dominating the development of the instability. Increasing the resolution to $C96$, $C192$ or $C384$ reduces the impact of the grid imprinting, however the instability has still developed more than in the reference solution of Galewsky et al. (2004) at $T341$ resolution. This is consistent with the finite-element cubed sphere model of Thuburn and Cotter (2015) which uses a similar mixed finite element discretisation to the one used here. In Thuburn and Cotter (2015) it is stated that the grid imprinting may be exaggerated by the highly unstable initial state of this test.





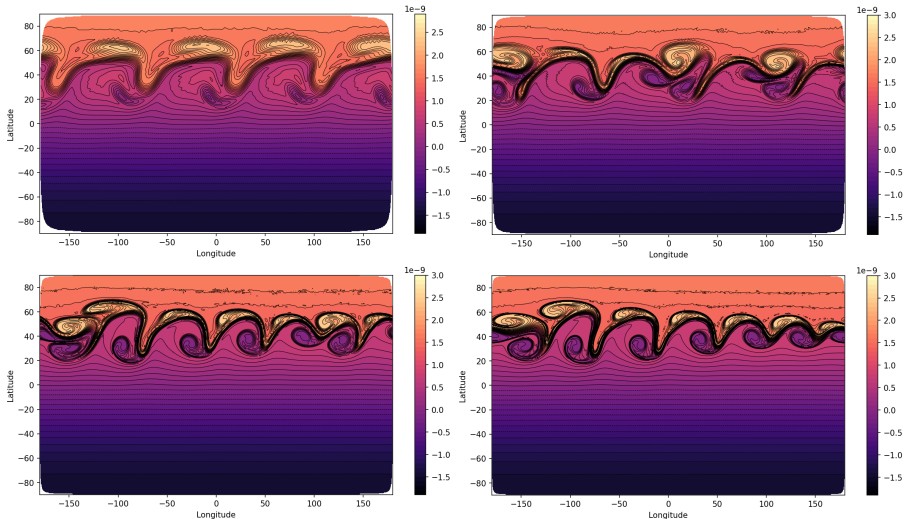

**Figure 6.** The potential vorticity $q$ after 6 days for the Galewsky instability test. The top left plot shows the $C48$ solution, the top right plot shows the $C96$ solution, the bottom left plot shows the $C192$ solution, and the bottom right plot shows the $C384$ solution.

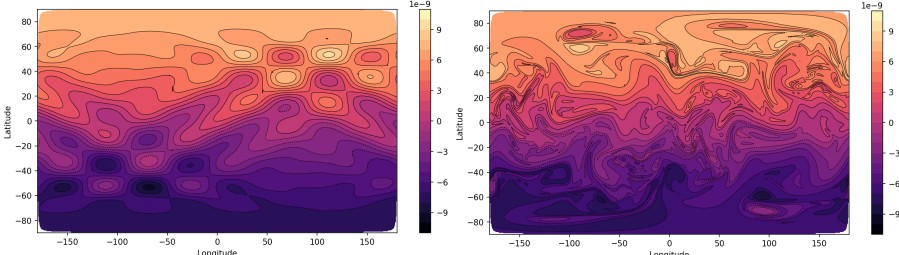

**Figure 7.** The potential vorticity $q$ for the vortices test case. The left plot shows the initial conditions, and the right plot shows the solution after 15 days from a high-resolution $C192$ simulation.

At the higher resolution many small scale features are resolved by the model without producing grid scale noise. This is due to the implicit diffusion from the transport scheme damping grid scale features. These solutions are comparable to the results shown in Thuburn et al. (2014) and Thuburn and Cotter (2015).

### 7.4 Vortices Test

The final test case is the field of vortices test from Kent et al. (2016). A field of vortices (shown by the potential vorticity in the left hand plot of Figure 7) is set to evolve over a number of days. The vortices interact, leading to small scale vorticity filaments and fingers as the vortices are stretched out, as well as the merger of vortices. A high resolution run (using $C192$ resolution, with potential vorticity shown after 15 days in the right plot of Figure 7) is used as a reference solution.



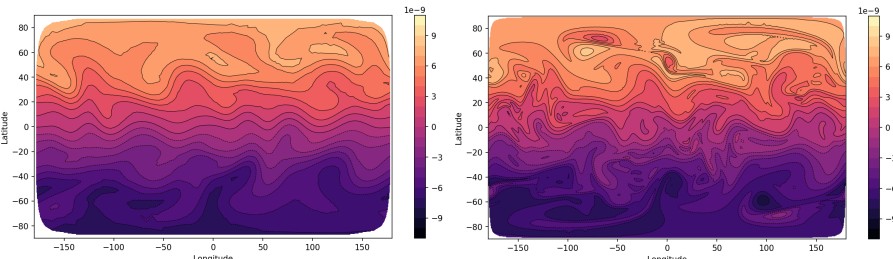

**Figure 8.** The potential vorticity $q$ for the vortices test case after 15 days for the $C24$ resolution (left) and the $C96$ resolution (right).

Figure 8 shows the day 15 potential vorticity for the $C24$ and $C96$ runs (with $\Delta t = 1800$ and $450$ s respectively). The $C24$ run is unable to resolve many of the features of the vortices, but the implicit diffusion from the transport scheme is
sufficient to prevent grid scale noise. For the higher-resolution $C96$ run, the small scale features of the potential vorticity are well represented, and the solution matches the reference solution well. This demonstrates the model's ability to represent small scale features without producing grid scale noise.

## 8 Conclusions

This article presents a new shallow water model on the sphere. The model is comprised of a mixed finite-element spatial
discretisation, a high-order finite-volume transport scheme, and an iterated semi-implicit time scheme, and makes use of the cubed sphere grid. The finite-element discretisation is chosen as it has been shown to be both accurate and scalable on many processors (Cotter and Shipton, 2012). The lowest-order finite-element method is used to give second-order spatial accuracy, and the finite-element spaces are such that they are analogous to a C-grid staggering. The semi-implicit time stepping provides stability for the fast gravity waves along with second-order temporal accuracy. The method-of-lines advection scheme, using
third-order Runge-Kutta time stepping with a quadratic finite-volume reconstruction, gives high accuracy and conservation for transport. The cubed sphere grid removes the pole problem of the latitude-longitude grid, and thus is more scalable (Staniforth and Thuburn, 2012).

The model has been tested on a standard set of shallow water test cases. The Williamson steady state test shows that the model converges at second-order accuracy. The mountain test demonstrates the model's ability to capture flow over orography.
Grid imprinting from the Galewsky instabilty test is evident at coarse spatial resolutions, but is significantly reduced at higher resolutions in line with results from other models with similar discretisations and cubed sphere meshes. The vortex field test highlights that the model can resolve small scale features without producing grid scale noise. The results presented from this testing are comparable to other models in the literature. This indicates that this model has a similar level of accuracy to these other well known shallow water models.



This shallow water model has been developed alongside the mixed finite-element Cartesian model for atmospheric dynamics of Melvin et al. (2019). The vector-invariant model presented here is a building block towards a mixed finite-element spherical geometry dynamical core for the atmosphere.

*Code availability.*    The shallow water model code can be found at https://github.com/thomasmelvin/gungho-swe. This repository includes the configuration files used for the tests in this article, and instructions for running the model.

The model code and configuration files are also freely available from the Met Office Science Repository Service (https://code.metoffice.gov.uk/ last access: 11 July 2022) upon registration and completion of a software licence.

*Author contributions.*    JK and TM developed the shallow water code with contributions from GW. JK performed the model simulations. JK and TM prepared the manuscript with contributions from GW.

*Competing interests.*    The authors declare that they have no conflict of interest.

*Acknowledgements.*



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
