# Peer review of "A mixed finite element discretisation of the shallow water equations"

_Geoscientific Model Development, 2022_

## Author Response (AR1)

**Response to Reviewers: A mixed finite element discretisation of the shallow water equations**

James Kent[1], Thomas Melvin[1], and Golo Wimmer[2]

[1]Dynamics Research, Met Office, Exeter, UK
[2]Los Alamos National Laboratory, Los Alamos, New Mexico 87545, USA

**Correspondence:** James Kent, Met Office, FitzRoy Road, Exeter EX1 3PB, UK. (james.kent@metoffice.gov.uk)

**1 Editor Comments**

Unfortunately, after checking your manuscript, it has come to our attention that it does not comply with our "Code and Data Policy".

You have archived the code of your shallow water model on GitHub. However, GitHub is not a suitable repository. GitHub itself instructs authors to use other long-term archival and publishing alternatives, such as Zenodo. Therefore, please, publish your code in one of the appropriate repositories according to our policy, and reply to this comment with the relevant information (link and DOI) as soon as possible, as it should be available for the Discussions stage. Also, in a potential reviewed version of your manuscript, you must include the modified 'Code and Data Availability' section with the DOI of the new repository.

Moreover, for the MetOffice code and data, you must be much more specific. You must provide the exact version number for the code and its DOI. For the data, if it can not be stored outside the MetOffice servers, you should provide a DOI, too, to ensure that it is easy to identify and obtain.

**Response to Editors Comments**

We have published our code on Zenodo - please see the link (which is also the DOI):

https://doi.org/10.5281/zenodo.7446738

**Manuscript Changes Due to Editor Comments**

We have included this link and DOI (https://doi.org/10.5281/zenodo.7446738) in the Code and Data Availability section of the manuscript (line 340), along with the revision number of the code. We have kept the link to the GitHub page in the manuscript as this might be more useful for some readers.

**2 Reviewer 1**

**General Comments**

While mimetic methods are becoming more common, it seems a lot to ask of the interdisciplinary GMD audience to follow a discussion of function spaces and de Rham complexes without some assistance. An illustration such as Figure 4 from Melvin et al. (2018) that corresponds to the specific cases mentioned by equations (7) and (8) would be most helpful.

Similarly, the article could benefit from a quick reminder of why mixed finite element methods are useful, especially since the lowest order formulation is used here. Differences between finite element methods, finite volume methods, and finite difference methods often disappear when used with low order discretizations. What is gained here that would not be present in a staggered finite volume method such as the one presented by Thuburn et al. (2014)?

Given the emphasis given to computational efficiency in the introduction, I expected more discussion of the method's computational performance. Detailed scaling studies are likely unrealistic this stage of development, but some general discussion would be helpful. Are the expected gains strictly due to the choice of grid, i.e., cubed sphere vs. latitude-longitude? Or is the numerical method helpful, too, for example, are its stencils for field reconstruction (e.g., Figure 1) smaller than other methods, implying less communication is required during runtime?

**Response to Reviewer 1 General Comments**

Thank you very much for your review.

We have produced a figure (the new Figure 1) that shows the mixed finite element spaces used in the model and have included it in section 3.2.

We agree that at lowest order these types of methods often become very similar. However, in Thuburn and Cotter, JCP, 2015, it is shown that the lowest order FE discretization has more benefit when it comes to consistency of the Coriols on non-orthogonal meshes than the FV model of Thuburn et al 2014. Another benefit of FE is the flexibilty to go to a higher-order element model. We have included this discussion in the introduction of our manuscript (line 30).

We've added some discussion about computational efficiency to the conclusions (line 325). We highlight that the cubed sphere grid has fewer cells than a corresponding lat-lon grid, and that the cubed sphere removes the pole and associated issues with parallel computing. Regarding the stencil size, the MoL transport scheme uses a small stencil for each reconstruction, but it must compute a reconstruction for each stage of the RK scheme. It is not clear at this stage whether this improves communication cost when compared to a scheme with a large stencil that is only called once.

**Specific Comments**

1) Equations (5) and (6) suggest that the advecting velocity is $\overline{\boldsymbol{u}}^{1/2}$ even in cases where $\alpha \neq 1/2$; is this true? Wood et al. (2014) suggest that off-centering by setting $\alpha > 1/2$ is important in the context of a 3D deep atmosphere model with orography. Is that concern relevant here, given that the method is presented as a step toward a full 3d atmosphere model?

2) How many iterations of GMRES are typically required to solve (33)? How sensitive is this number to the resolution?

3) The description in Section 6 of a "finite-element representation of the sphere within a cell with polynomials" is difficult to follow. I assume that the four vertices of an element lie on the sphere; for the case of a quadratic elements, are the nodes that are not vertices also on the sphere?

4) Is the fact that some internal points of a cell may not exactly lie on the spherical surface related to the fact that different function spaces are used for different variables? It doesn't seem to be an issue with other finite-element dynamical cores such as Guba et al. (2014), that also rely on mappings to and from a reference quadrilateral.

5) I found the discussion of error at the beginning of Section 7.1 confusing; it states that the method is second-order in both space and time, but immediately preceding this remark at the end of Section 6, fourth-order convergence is cited as the reason for choosing quadratic elements. I agree that the method should be overall second order, so I presume that the fourth-order convergence refers to reconstructing the spherical surface itself, rather than an arbitrary scalar function?

**Response to Reviewer 1 Specific Comments**

1) This is true, we use $\overline{\boldsymbol{u}}^{1/2}$ even if alpha $\neq 1/2$. This is consistent with Wood et al. 2014, and is used to get the second-order time discretization. Currently we use $\alpha = 0.5$ in the model configuration. For shallow water we have not seen the need to off-centre. We agree however that this is important for a full 3D model.

2) It seems to take around 2-3 iterations for GMRES to converge to a tolerance of $10^{-4}$ on the C24 and C48 grids for both the mountain and Galewsky test. We have stated this at the end of section 5 (line 218).

3) We have rewritten parts of section 6 to make things clearer. We've described a linear element (line 223) and explained why we use the mesh parameterisation (line 224). You are correct that for the quadratic elements all the nodes lie on the sphere, and quadratic functions represent the sphere between these nodes.

4) Representing the sphere with elements removes the need for analytic transformations (we state this on line 224). This means we can use arbitrary grid if we want, although in this paper we only consider the cubed shpere grid. A down side is we are parameterizing the sphere, but as shown using quadratic elements on the C96 grid gives a maximum error of 0.0018 m. The parameterisation of the sphere is not related to the function spaces used.

5) You are correct that the fourth-order is for the spherical surface, and the second-order is for the Williamson 2 test case. We've added line 233 to make this clearer.

**3 Reviewer 2**

**Reviewer 2 Specific Comments**

The motivation for this new model could be a lot stronger. Much of the motivation provided could have been written last century, for example the need for parallisation and the need to go beyond finite differences, finite volume and semi-Lagrangian. The motivation for mixed finite elements is easy and has already been written about. The motivation needs in involve massive parallelisation, wave dispersion, spectral elements and DG.

Section 4 needs to define the order of accuracy in space of the transport scheme. I think it must be limited to two because you do not define how you fit a polynomial using cell average values.

Figure 4 and the related discussion (lines 239-243) are weak. Figure 4 only really shows that your model works. It doesn't, as you say, show that the "results are comparable to other shallow water models" or demonstrate "the model's ability to correctly simulate flow over orography". I would plot errors rather than figure 4 (in comparison to STSWM) and convergence with resolution. It is also informative to show the vorticity after 50 days which is a good indicator of conservation, balance and a lack of spurious artefacts in the solution. Eg see:

Fig 11 of "A unified approach to energy conservation and potential vorticity dynamics for arbitrarily-structured C-grids", Journal of Computational Physics 229 (2010) 3065–3090

or fig 5 of

"Computational Modes and Grid Imprinting on Five Quasi-Uniform Spherical C-Grids", Weller, Thuburn and Cotter.

**Response to Reviewer 2 Specific Comments**

Thank you for reviewing our paper.

Regarding motivation, we have rewritten parts of the introduction to takes these points into account. We have stressed the need for massively parallel models for the future of weather and climate forecasting, and have discussed the benefits of mixed finite-element methods over, for example, finite-volume methods (line 29).

The transport scheme in section 4 is actually 3rd order in space and time. The temporal order comes from the SSPRK3 algorithm. The spatial order comes from the quadratic reconstruction of the field at flux points. The fitting of the polynomial is such that the integral of the polynomial is equal to the integral of the variable within each cell. We have made this clearer in

the text in section 4 (lines 126 and 135).

125  We have significantly rewritten large parts of the mountain test case section. We use a high-resolution semi-implicit semi-Lagrangian scheme as a reference to produce error plots, which we then compare with other models in the literature (line 255). We also look at the error convergence with resolution (lines 265 and 268). We have extended the energy and potential enstrophy statistics to 50 days, and provided a plot of the day 50 potential vorticity (discussed in the manuscript at line 288).

**Reviewer 2 Technical Corrections**

130  Try to make your writing more concise. For example, delete phrases like "and the interested reader is referred there for more information".

In table 1, use scientific notation rather than exponents.

**Response to Reviewer 2 Technical Corrections**

135  We have removed the text "and the interested reader is referred there for more information" (lines 56 and 130) and have used scientific notation in the error norm table (Table 1).